# Prebiotic Effect of Oxidized Hydroxypropyl Starch via In Vitro and In Vivo

**DOI:** 10.3390/foods14132217

**Published:** 2025-06-24

**Authors:** Huiwen Zheng, Zhipu Xu, Yiwen Fan, Jiazhi Han, Liyang Zhou, Han Li, Xiaohua Pan, Rongrong Ma, Chang Liu, Yaoqi Tian

**Affiliations:** 1State Key Laboratory of Food Science and Resources, Jiangnan University, Wuxi 214122, China; 2School of Food Science and Technology, Jiangnan University, Wuxi 214122, China; 3School of Internet of Things Engineering, Wuxi 214122, China; 4Analysis and Testing Center, Jiangnan University, Wuxi 214122, China

**Keywords:** oxidized hydroxypropyl starch, digestion, ferment, gut probiotics

## Abstract

Most studies on resistant starch are limited to its effect on blood glucose; there are few studies on the prebiotic effects of resistant starch on the gut. In this experiment, through in vivo metabolism verification and in vitro simulated fermentation experiments, it was found that hydroxypropyl oxide (OHS) had a prebiotic effect on the intestine. The results of bioinformatics showed that the structure of the microbiota changed significantly, and the in vitro and in vivo fermentation results of *Bacteroides uniformis* and *Parabacteroides distasonis* showed an upward trend. The results of a KEGG prediction of the metabolic pathway showed that Phenylalanine metabolism and Cysteine and methionine metabolism showed an enhanced trend. At the same time, the results of in vitro and in vivo metabolite assays further confirmed this point, and the content of L-Homocystine and Phenylalanine in metabolites decreased significantly, with the decrease in L-Homocystine posing a reduction in cardiovascular disease risk and the decrease in Phenylalanine having a positive significance for phenylketonuria patients. This study proved that hydroxypropyl oxide can regulate the intestinal microbiota and has intestinal prebiotic effects, which can be used to guide the development of functional foods.

## 1. Introduction

Starch is a polysaccharide found in plants with a structure consisting mainly of amylose and amylopectin, which is mainly used as a source of energy [1]. However, the natural properties of natural starch give it certain limitations that may not meet the needs of the food industry. Therefore, the modification of natural starch to obtain resistant starch with specific functionalities, which enhances the starch’s advantages while eliminating possible limitations, can enable the starch to further meet the needs of the modern industry [2]. Modified starches are mainly categorized into RS1 (physically encapsulated), RS2 (high straight-chain starch), RS3 (regenerated starch), and RS4 (chemically modified starch). At present, there is still a partial gap in the research on hydroxypropyl starch and oxidized hydroxypropyl starch in RS4.

The bioavailability of starch in the small intestine can be categorized into rapidly digestible starch (RDS), slowly digestible starch (SDS), and resistant starch (RS) with anti-digestive properties. Resistant starch is difficult to digest and is not utilized in the small intestine. It can reach the colon, where it is utilized by microorganisms in the intestine to produce short-chain fatty acids, which have a prebiotic effect on the intestinal tract [3]. Studies have shown that resistant starch has intestinal prebiotic effects and positively affects the growth of *Bifidobacteria* and anabolic bacteria [4] and modified oat starch reduces insulin resistance, inhibits the release of inflammatory factors, and modulates the gut microbiota [5]. The human gut is a complex ecosystem in which microbial diversity, metabolic flexibility, functional redundancy, and host–microbiota interactions are all important for maintaining gut microbiota homeostasis [6]. The gut microbiota is involved in a variety of metabolic functions, and one of its core functions is to participate in nutrient catabolism, including undigested carbohydrates [7]. A healthy gut microbiota is essential for promoting host health [8]. Therefore, it is worthwhile to explore the intestinal prebiotic effects of resistant starch to further develop its functions. However, there are limitations in the study of hydroxypropyl and oxidized hydroxypropyl at this stage, and their prebiotic effects on the intestinal tract are not clear.

In vitro fermentation experiments by collecting human feces and placing them under the appropriate temperature and pH for fermentation are easy and fast, and there are no ethical constraints, but they cannot fully simulate the changes in the microbiota and metabolites in the intestines of real individuals. In vivo digestion can more realistically simulate the process of starch changes in real individuals by examining the microbiota and metabolites in mouse feces. Combining in vitro and in vivo fermentation to explore the changes in intestinal microorganisms and metabolites during the fermentation process can make the simulation more realistic and reliable, reduce the consumption of experimental animals, and realize the advantages of complementarity.

It has been shown that modified starches may have potential in antidigestive function and gut prebiotics [9], while studies on the role of etherified starches in these two aspects remain unclear. The aim of this investigation is: (1) to investigate the changes in the properties of hydroxypropyl starch (HPS) and oxidized hydroxypropyl starch (OHS) relative to common corn starch (CCS); (2) to explore the possible antidigestive properties of HPS and OHS using in vitro simulated digestion experiments; (3) to explore the probiotic effects of HPS and OHS on the intestinal tract; and (4) to fill the gap in the research on modified starch RS4. In this study, the structural properties and crystal structures of the prepared starches were determined, and the antidigestive properties of OHS and HPS were explored by in vitro digestion. The potential probiotic effects of HPS and OHS were then identified from both in vivo and in vitro perspectives through in vivo and in vitro metabolomics analysis of the fermentation products. This study will provide a basis for further research on starch.

## 2. Materials and Methods

### 2.1. Experimental Materials

Common corn starch (CCS) provided by Hangzhou Starpro Starch Co., Ltd, Hangzhou, China, amyloglucosidase (enzyme activity ≥ 260 U/mL, A7095, EC 3.2.1.3), and porcine pancreatin (P7545, EC 232-468-9, 8USP) were purchased from Sigma–Aldrich; the glucose oxidase (GOPOD) kit was provided by Beijing Liedman Biochemical Co., Ltd., Beijing, China; Heparin sodium, metaphosphoric acid, crotonic acid, yeast extract, tryptone, potassium dihydrogen phosphate, L-cysteine, and other pure reagents required for analysis were provided by Sinopharm Chemical Reagent Co., Ltd., Beijing, China.

### 2.2. Experimental Methods

#### 2.2.1. Preparation of Hydroxypropyl Starch (HPS)

Hydroxypropyl starch was prepared by referring to the method of Park et al. [10] with slight modification. Corn starch (36% *w*/*w*), anhydrous sodium sulfate (12% *w*/*w* of corn starch), and deionized water were mixed at 40 °C. The pH (measured by Precision pH-Meter: FE28-Standard, Mettler Toledo Instruments, Zürich, Switzerland) was adjusted to 11.5 with 0.1 M NaOH solution; propylene oxide (12% *w*/*w*) was added to the dispersion system and transferred to a constant temperature water bath (HZQ-2 type, Changzhou Guowang Instrument Co., Changzhou, China) at 40 °C with continuous stirring for 20 h. After the reaction was completed, the slurry was adjusted to pH 5.5 with 0.1 M HCl and vacuum filtered. The slurry was resuspended with distilled water and filtered again, washed three times, dried in an oven (Model 101, Shanghai Keheng Industrial Development Co., Ltd., Shanghai, China) at 40 °C for 24 h, and then sealed and stored after grinding.

#### 2.2.2. Preparation of Oxidized Hydroxypropyl Starch (OHS)

Oxidized hydroxypropyl starch was prepared by the further oxidation of hydroxypropyl starch, and the preparation method was referred to Zhang et al. [11]. The prepared hydroxypropyl starch was dissolved into a 40% (*w*/*w*) suspension with deionized water and stirred thoroughly. The whole reaction was conducted at 40 °C in a water bath (HZQ-2 type, Changzhou Guowang Instrument Co., China). The solution pH was adjusted to 8.0 with 4% (*w*/*w*) NaOH solution, and then sodium hypochlorite solution (5% effective chlorine content *w*/*w*) was added. The reaction was carried out with 0.1 M H_2_SO_4_ for 3 h at a controlled pH of 9. After completion, the pH was adjusted to 7 with 0.1 M H_2_SO_4_, filtered under vacuum and washed three times with deionized water. It was dried in an oven at 40 °C (Model 101, Shanghai Keheng Industrial Development Co., Ltd., China) for 24 h. After grinding, it was sealed and stored.

#### 2.2.3. Crystalline Structure Determination

X-ray diffraction (XRD) was used to determine the crystalline structure of the sample for analysis. The powdered samples were first spread evenly in the center of the circular groove of the sample stage, and a slide was used to compact and flatten the sample surface. The sample stage was then placed in an X-ray diffractometer (D2 PHASER model, Bruker AXS, Karlsruhe, Germany). The test conditions were as follows: a Cu-Kα radiation light source of 40 kV and 30 mA was set, the scanning range was set at 4–40° (2θ), and the scanning rate was 4° per min. The relative crystallinity of the samples was calculated by the Jade 6.0 software using the formula:Rc = Ac/(Ac + Aa) × 100%
where Rc is the relative crystallinity of the sample, Ac denotes the area of the crystalline zone of the sample, and Aa denotes the area of the amorphous zone of the sample.

#### 2.2.4. FT-IR Spectra Determination

The short-range ordered structure of the sample starch was determined by a Fourier infrared spectrometer (IS10, Thermo Nicolet Corporation, Madison, WI, USA). Air was pre-collected as a background, and powdered samples were taken until they were exactly flat on the sample stage of the ATR-FTIR to map the infrared spectra. The acquisition range was set from 600 to 4000 cm^−1^ with 32 scans, and the resolution was set to 4 cm^−1^.

#### 2.2.5. Measurement of Changes in Thermodynamic Properties

The thermodynamic properties were determined by a differential scanning calorimeter (DSC3 model, Mettler-Toledo International Inc., Zurich, Switzerland). 4.0 mg of sample starch was weighed with an analytical balance and placed in an aluminum crucible, ultrapure water was added at a mass ratio of 1:2, and the lid of the sealed ceramic crucible was compacted using a DSC press and left overnight to equilibrate the water. The instrument was calibrated with indium standards before the DSC test. After calibrating the instrument, the sealed empty aluminum crucible was used as a blank control, the heating temperature was set at 25–125 °C, the heating rate was set at 10 °C/min, and the flow rate of nitrogen (N2) was adjusted to 20 mL/min, and the heat flux profile was analyzed using the DSC on-board software (DSC3, STARe Evaluation Software 12.1).

#### 2.2.6. Determination of Carboxyl Group Content

The carboxyl groups were determined by reference to the titration method of Sares et al. [12]. 2 g of sample was weighed and suspended in 25 mL of 0.1 M HCl solution and stirred intermittently with a magnetic stirrer for 30 min. Vacuum filtration was carried out, the sample was washed with 400 mL of distilled water, the filter cake was dissolved in distilled water, and the volume was adjusted to 300 mL. The starch suspension was placed in a boiling water bath for 15 min, during which stirring was maintained to ensure complete pasting. The volume was adjusted to 450 mL with distilled water, and the pH was titrated to 8.3 with 0.01 M NaOH solution. The carboxyl group content was calculated as follows:Carboxyl contents = [(Sample − Blank) × Molarity of NaOH]/Sample weight × 100
where CCS was used as Blank, Molarity of NaOH = 0.01 M

#### 2.2.7. In Vitro Digestion Characterization

The in vitro digestive properties of sample starch were determined by the method of Englyst [13] with some modifications. 4.5 g of porcine pancreatic enzyme was mixed with 40 mL of ultrapure water, vortexed for 5 min until it was uniformly dispersed, and centrifuged (GL-10MD, Hunan Xiangyi Centrifuge Instrument Co., Changsha, China) at 5000 rpm for 10 min. 27 mL of the supernatant was taken from the centrifuge tube, and 3.2 mL of amyloglucosidase (≥260 U/mL) was added and then vortexed and oscillated for 5 min to obtain the enzyme-mixed solution. A 0.5 mol/L sodium acetate (pH 5.2) buffer ethanol aqueous solution (90% *v*/*v*) was prepared. A total of 0.2 g of starch was weighed in a centrifuge tube, 4 mL of ultrapure water was added to dissolve, and vortexing and shaking were carried out to make a homogeneous mixture. Eight glass beads and 4 mL of sodium acetate buffer were added, and the mixture placed in a boiling water bath for 15 min. Subsequently, 2 mL of enzyme mixing solution was added to the above centrifuge tubes, which were in the 37 °C thermostatic water bath shaker (SHZ-88 type, Jintan Shantou Linfeng Experimental Instrument Factory, Jintan, China) set at 150 r/min; timing was started after the enzyme mixing solution had been added to the above centrifuge tubes. At 0, 10, 20, 30, 60, 90, 120, and 180 min, 100 μL of solution was transferred to the centrifuge tube with 900 μL of aqueous ethanol (90% *v*/*v*), and centrifugation was performed at 10,000 rpm for 5 min. The glucose content of the supernatant was determined by the GOPOD method: 6 μL of supernatant was taken from the centrifuge tube (in three parallel portions) and added to the enzyme plate, 200 μL of the assay reagent was added to the plate, and 520 nm was measured using an Enzyme-Labeled Instrument (Power Wave Model XS2, BioTek Instruments, Inc., Winooski, VT, USA). The glucose contents G0, G20, and G120 were obtained from the decomposition of the samples at 0, 20, and 120 min. It was calculated using the formula:RDS (%) = (G20 − G0) × 0.9 × 100/XSDS (%) = (G120 − G20) × 0.9 × 100/XRS (%) = 1 − RDS (%) − SDS (%)
where X is the mass of starch sample taken (g) and 0.9 is the stoichiometric constant of starch in glucose.

#### 2.2.8. In Vitro Fecal Fermentation Experiments

The in vitro simulated fermentation experiments were carried out according to the method of Liu et al. [14] with some modifications. The culture medium was prepared according to the ratio, autoclaved at 121 °C for 20 min, and then removed and set aside. Three healthy volunteers were selected in order to provide fresh fecal samples (BMI qualified, no digestive diseases, no history of antibiotic or probiotic administration within 3 months). 0.8 g of feces was taken into the sample box, which was placed on the F100 fecal processor and processed by adding 8 mL of fecal diluent. The fecal suspension was made by vortex mixing, and the fecal suspension was filtered through a 100-mesh fecal filter sieve to remove large food debris to obtain the fecal filtrate. Starch samples were taken to simulate fermentation in vitro using human feces. Transfer 500 μL of fecal filtrate from the sample box to a syringe bottle, add 20 mg of starch sample and 5 mL of sterilized blank medium, mix well, and then put into a sterile anaerobic bag, and incubate at 37 °C for 24 h. After incubation, take out of the bag and put it into a refrigerator at −80 °C to be measured.

#### 2.2.9. In Vivo Enzymolysis of Starch

A total of 10 SPF-grade C57BL/6J mice (6–8 weeks old, 20 ± 2 g) were purchased from Zhejiang Viton Lihua Laboratory Animal Technology Co., Jiaxing, China. The protocol of experimental animals was approved by Jiangnan University Laboratory Animal Management and Animal Welfare Ethics Committee (JN.20250331c0150520[146]), and the animals were kept in the SPF class rearing room of Jiangnan University Laboratory Animal Centre (License No. SYXK(SU)2021-0056). The temperature of the room was maintained at 22~25 °C, and the humidity at 40~70%; the light/dark cycle was 12 h. The bedding, feed, and drinking water were changed regularly. The mice were healthy with no obvious abnormalities during the experimental period. Ten mice were divided into two groups according to their body weights, and the mice were gavaged with two kinds of starch (CCS and OHS, 15%, *w*/*v*, 0.25 mL) for 3~5 days after 16 h of fasting without water. After the mice were executed by carbon dioxide, the mice were dissected, and the desired colon and cecum were removed; the contents were separated and collected and placed in centrifuge tubes at −80 °C in a refrigerator for measurement.

#### 2.2.10. Analysis of Metabolites by GC-MS and LC-MS Measurements

Sample extraction and detection methods referred to Shi [15] et al. 100 μL of sample was pipetted into a centrifuge tube, and 200 μL of methanol solution containing 1 μg/mL of internal standard was added and vortexed for 10 min to make a homogeneous mixture. The extract was centrifuged (20,000× *g*, 4 °C) for 10 min, the precipitate was discarded, and the supernatant was dried under vacuum. The extract was completely dissolved in 200 μL of methanol and diffused by sonication under ice bath conditions for 15 min. The metabolites were analyzed by GC-MS (QP2010, Shimadzu Co., Columbia, MD, USA, and Triple TOF 5600, SCIEX, Scientific Export Inc., Redwood City, CA, USA) and LC-MS (ExionLC AD, SCIEX, Scientific Export Inc., Redwood City, CA, USA) determination.

For GC-MS, the inlet, ion source, and interface temperatures were set at 250 °C, 230 °C, and 280 °C. The column temperature was initially set at 70 °C, then increased to 250 °C at 10 °C/min and rapidly ramped up to 300 °C and held for 5 min. The carrier gas was helium at a flow rate of 1.0 mL/min.

For LC-MS, Liquid chromatography was performed on a Kinetex C18 column (100 mm × 2.1 mm, 2.6 μm) and a BEH amide column (100 mm × 2.1 mm, 1.7 μm), respectively. The specific procedure referred to the description by Shi [15].

Referring to the data analysis method of Liu [16], the open-source software MSDIAL 4.0 was used for the identification of the characterized metabolites, and the pathway analysis of the characterized metabolites was performed using MetaboAnalyst 5.0.

#### 2.2.11. 16S rRNA Gene and Bioinformatics Analysis

The analyzing method referred to Liu et al. [16]. DNA in the samples was extracted using a genomic DNA purification kit (Shanghai Sangong Biotechnology Co., Ltd., China, Shanghai). Primer pairs were used (341 F, 5′-CCTAYGGGRBGCASCAG-3′; 806R, 5′-GGACTACHVGGGTWTCTAAT-3′) for genomic PCR amplification (CFX96, Bio-Rad, Bio-Rad Laboratories, Hercules, CA, USA). PCR products were recovered and purified and then quantified using QuantiFluor™-ST (Promega, Beijing, China, Biotech Co., Ltd., Beijing, China). Qualified sequencing libraries were sequenced using a NovaSeq sequencer. The double-ended sequences were spliced after removing the barcode and linker sequences. Sequence denoising and clustering were performed through the dada2 analysis process, and representative sequences and feature lists were exported for species annotation and visualization. Gene functional analysis was performed using the PICRUSt2 software v2.5.2 package, and metabolic pathway and KO enzyme gene analysis was performed based on the KEGG database (http://huttenhower.sph.harvard.edu/galaxy/, accessed on 13 February 2025).

#### 2.2.12. Data Processing and Analysis

All tests were repeated more than three times, and the results were expressed as mean ± standard deviation. Data were processed using the Prism 10 software and analyzed by ANOVA and Duncan’s multiple comparison test. *p*-value < 0.05 was considered statistically significant. Structural equation modeling analysis was performed using R-studio based on the chi-square test of non-significance (*p*-value > 0.05)

## 3. Results and Discussion

### 3.1. Preparation of Resistant Starch and Characterization of Its Structural Properties

#### 3.1.1. Crystal Structures and Pasting Properties

XRD was used to investigate the crystal structure of HPS and OHS made from CCS, and the XRD spectra of the three starches are shown in Figure 1A. Figure 1A shows that CCS, HPS, and CMS show characteristic diffraction peaks at 15°, 17°, 18°, and 23° (2θ), which are typical A-type crystal structures [17], suggesting that the reactions during the hydroxypropylation and oxidation of these two etherified starches were mainly carried out in the amorphous region, and thus did not completely disrupt the altered starch crystal shape. The crystallinity of the three starches was further calculated; the mean values of the crystallinity of CCS, HPS, and OHS were calculated to be 0.2836, 0.2395, and 0.2347, respectively, and the significance plots are shown in Figure 1B, which shows that the crystallinity of HPS and OHS was significantly reduced compared to that of CCS, indicating that the treatment process inevitably caused damage to the crystal type. It may be due to the embedding of the hydroxypropyl group during the modification process, resulting in partial hydrogen bonding disruption and a localized distortion of the double helix structure. In addition, the crystallinity of OHS decreased compared to HPS, but the change was not significant, which is consistent with the findings of Kuakpetoon et al. [18].

The melting characteristics of etherified starch were determined by the DSC method, and the results are shown in Figure 1C,D, where T_0_, T_P_, T_C_, and ΔH represent the onset melting temperature, peak temperature, conclusion temperature, and enthalpy of melting, respectively [19,20]. The results of Figure 1C show that the peaks of OHS and HPS are shifted to lower temperatures and the peak area is reduced compared to CCS. Figure 1D also shows that the T_0_, T_P_, T_C_, and ΔH of etherified starch were lower than those of CCS, where T_0_, T_P_, and T_C_ were decreased by 10~15 °C and ΔH was decreased by about 5~10 J/g. The decrease in the pasting temperature may be due to the disruption of hydrogen bonding between the starch chains in the amorphous region of etherified starch, which increases the mobility of the starch chains and indirectly decreases the melting temperature of the starch microparticles [21], which is in agreement with the results of Woggum’s study [22]. The decrease in ΔH may be the result of the ether bonding introduced to weaken the double helix structure, leading to a decrease in crystallinity [17], which is consistent with the XRD results. In addition, the melting temperature of OHS was slightly higher than that of HPS, and ΔH was slightly lower than that of HPS, as shown in Figure 1D. The increase in melting temperature may be due to the hydrolysis of the amorphous zone due to oxidation, leading to the increase in pasting temperature [23], while the decrease in ΔH may be due to the weakening of starch granules due to the further degradation of starch molecules as a result of oxidation [24], which was also in accordance with the results of the previous XRD analyses.

#### 3.1.2. FT-IR Spectra and In Vitro Digestion Results Analysis

The short-range ordered structures of the three starches were analyzed using FT-IR (shown in Figure 2A). In the spectrum near 3400 cm^−1^, the broad peak is generated by O-H stretching vibration, and the short peak at 2930 cm^−1^ is generated by C-H stretching vibration, and they are the characteristic absorption peaks of polysaccharides. In addition, the characteristic peak located at 1640 cm^−1^ is related to scissors vibrations of O-H coming from the water of hydration [25]. The characteristic peaks near 1160 cm^−1^, near 1100 cm^−1^, and near 1000 cm^−1^ are related to the stretching vibrations of glycosidic bonds to C-O-C and C-O-H, which are also characteristic of polysaccharides [25]. In the FTIR pattern of HPS, the enhanced peak appearing near 2970 cm^−1^ [26] is caused by the stretching of the asymmetric CH(CH_3_) of the hydroxypropyl species, which can be taken as a sign of the success of hydroxypropyl preparation. In addition, from the infrared map of OHS, it can be found that the bending vibrational absorption peak of OHS near 1640 cm^−1^ is shifted toward 1600 cm^−1^ compared to HPS [11], which is consistent with Zhang’s study and proves that the preparation of OHS starch was successful. This is further supported by the research of Ye et al. [27]. The carboxyl content of OHS was further calculated. It was noted that both Carbonyl content and Carboxyl content can be used for the validation of oxidatively modified starch [28] but Carboxyl content tends to be more significant. Therefore, Carboxyl content was chosen for the further validation of oxidized hydroxypropyl starch. The results are shown in Table 1, and it is clear that the carboxyl content in the hydroxypropyl starch is negligible while the further oxidation successfully introduces carboxyl groups. The Carboxyl content of the resulting oxidized hydroxypropyl starch was about 0.56%, a result that is in agreement with the study of Zhang et al. [11], i.e., when the effective chlorine is 5%, the Carboxyl content is about 0.6%. Combined with the infrared absorption peak shift, this further confirms that the oxidized hydroxypropyl modification was successful and is of moderate oxidation.

The distribution of starch composition (RDS/SDS/RS) of the three starches after cooking is shown in Figure 2B. Both hydroxypropyl and oxidized hydroxypropyl modification reduced the RDS value of the starch from 85% to about 50% and increased the RS value from less than 2% to about 23% and 26%, respectively. This indicates that the modified starch has some resistance to enzymatic degradation and more of the undigested portion [13] can enter the large intestine to be utilized by microbial fermentation. The substitution of a hydroxypropyl group can increase the enzyme resistance of starch, but it is affected by multiple factors such as original starch properties, degree of substitution, and branched chain spatial resistance [29]. Further oxidative modification did not result in significant results, resulting in only a small increase in RS content, probably because although oxidation introduces the carboxyl group, which has better resistance, it also destroys part of the hydroxypropyl group by oxidation [10], thus counteracting part of the enzymatic resistance.

### 3.2. Analysis of Intestinal Microbiota Results

In vitro fermentation can be used to evaluate the ability of human gut microorganisms to metabolize food through gut microbiota and an anaerobic environment. The Simpson and Shannon indices were chosen to characterize the results, which are important reference indices used to measure biodiversity, mainly reflecting the degree of dominance of the species in the community. The larger values of the Simpson and Shannon indices indicate that the species are uniformly distributed and that the diversity is higher [30]. The results of α-diversity analysis (Figure 3A) showed that the Simpson and Shannon indices of HPS and OHS decreased after in vitro fermentation compared with those of CCS, indicating that HPS and OHS reduced the diversity of the tract microbiota to different degrees. The β-diversity analysis of the microorganisms after in vitro fermentation was further performed, and the principal component analysis (PCA) is shown in Figure 3C. The microbial compositions of HPS and OHS were further away from that of CCS, which further indicated that HPS and OHS affected the microbial community composition.

The modification of the microbial community was quantified at the phylum level by 16S rRNA gene sequencing, and the measurements were shown in Figure 3E. CCS, HPS, and OHS were mainly composed of Bacteroidota, Firmicutes, Actinobacteriota, and Proteobacteria, with a relative abundance of at least 95%. The abundance of Bacteroidota and Firmicutes was significantly higher, and Actinobacteriota was significantly lower in the HPS group and OHS group compared to the CCS group. Studies have shown that *Bacteroidota* is a diverse bacterial phylum that is increasingly recognized for its significant contribution to host health, especially through its antimicrobial and probiotic properties [31]. In addition, Firmicutes were shown to produce more butyrate. Butyrate is considered a health-promoting molecule due to its ability to increase insulin sensitivity, exert anti-inflammatory activity, regulate energy metabolism, and increase leptin gene expression [32]. Another study showed a positive correlation between *Actinobacteria* and BMI at the phylum level and that Actinobacteria may cause obesity [33]. To further investigate the effect of etherified starch on human gut microbiota, 16S rRNA gene sequencing was then used to quantify the modification of microbial communities at the species level.

Figure 3G demonstrates the variation in the abundance of microbial communities at the species level. LDA (Figure 3I) showed that the HPS group, OHS group, and CCS group differed significantly on 10 species. Further significance analysis (Figure 3K) revealed that the relative abundance of *Bacteroides uniformis* in the HPS and OHS groups was significantly higher (*p* < 0.01) compared with that of the CCS group. It has been reported that *Bacteroides uniformis* was able to degrade β-glucan and promote *Lactobacillus johnsonii* to alleviate colitis by increasing indole-3-lactic acid levels [34]. This bacterium has also been shown to have endurance-enhancing properties [35]. In addition, the abundance of *Parabacteroides distasonis* was significantly increased (*p* < 0.01), and the study suggests that *Parabacteroides distasonis* has an anti-inflammatory effect in the intestinal microbiome and is capable of exerting intestinal epithelial monolayer-enhancing effects to strengthen the intestinal barrier [36]. Furthermore, HPS and OHS led to a significant increase in the abundance of *Agathobacter rectalis*, and recent studies have shown that the proliferation of this bacterium ameliorates the pathological damage caused by Alzheimer’s disease (AD) and exerts an inhibitory effect on microglia activation [37].

The relative abundance of *Dialister hominis* was significantly increased (*p* < 0.05), and studies have shown that circulating succinate levels are higher in obese patients. Huber-Ruano et al. recently reported that *D. hominis* is one of the candidates for reducing circulating succinate and ameliorating obesity-related inflammation in obese patients [38]. The above findings imply that HPS and OHS can add value to the beneficial microbiota and optimize the gut microbial structure, e.g., *Bacteroides uniformis*, *Dialister hominis*, *Parabacteroides distasonis,* etc., which are important in the regulation of the gut microbiota, maintenance of intestinal health, anti-inflammatory, and so on.

HPS and OHS showed more consistent effects on gut microorganisms in in vitro simulated fermentations, both resulting in changes in microbial diversity and a proliferation of beneficial microbiota at the phylum level and species level. Comparing the results of the two starches, OHS appeared to be more significant in diversity effects and proliferation of probiotics, and, combined with the previous section, which showed that the RS content in OHS was higher than that of HPS, the synthesis can show that OHS was slightly superior to HPS; and therefore, OHS was chosen to further carry out the in vivo fermentation experiments in vivo.

In order to further prove the role of etherified modified starch represented by OHS in regulating intestinal microorganisms, the intestinal microbiota of mice was investigated by in vivo fermentation in mice. The results of α-diversity determination are shown in Figure 3B. Compared with the CCS group, the Simpson value decreased, which is consistent with in vitro fermentation. However, the Shannon value increased, which is different from the results of in vitro fermentation. It may be because the microbial community in the intestine of mice is affected by various factors such as the host physiological state, immune system, and external environment [39], and the situation is even more complicated. Similar to in vitro fermentation, the results of β-diversity analysis of in vivo fermentation are shown in Figure 3D. The results of the two groups are completely separated, further demonstrating that OHS has a significant effect on the composition of intestinal microbial communities.

Figure 3F shows the modification of microbial communities in vivo at the phylum level, similar to the results of fermented microbial communities in vitro. Compared with the CCS group, the abundance of Bacteroidota and Firmicutes in the OHS group increased significantly, and the abundance of Actinobacteroida decreased significantly, further demonstrating the regulatory effect of OHS on intestinal microbiota.

Figure 3H shows the modification of microbial communities in mice at the species level. The LDA results (Figure 3J) showed that there were significant differences in OHS among the 10 strains after fermentation. Further combined with the significance analysis results (Figure 3L), it can be found that the abundance of *Bacteroides uniformis* and *Parabacteroides distasonis* has increased significantly, which is consistent with the results of in vitro fermentation, demonstrating that OHS can play its potential beneficial role in relieving colitis and strengthening the intestinal barrier by regulating the intestinal microbial structure. The in vivo fermentation results also show that the number of *Bacteroides thetaiotaomicron* has increased significantly. The bacteria is widely distributed in the human intestine, and its role in the fermentation of polysaccharides has been widely investigated [40]. Porter et al. [41] pointed out that *Bacteroides thetaiotaomicron* has a powerful starch utilization system, while providing the host with absorbable short-chain acids and organic acids, with potential health properties. In addition, feeding OHS significantly reduced the abundance of *Acetatifactor* in vivo, which was confirmed to be positively correlated with obesity and diabetes [42], so reducing the abundance of this bacteria is obviously beneficial to health.

### 3.3. Predicting Functional Changes in the Microbiota

To further analyze how OHS-induced changes in intestinal microbiota regulate host metabolism, metagenome-based PICRUSt2 [16] prediction will be used to predict metabolic changes. The KEGG database based on is a comprehensive bioinformatics database covering genes, proteins, pathways, biochemical reactions, etc. It is often used in metabolomics analysis [43] and aims to predict the change trend of metabolic pathways [44]. By comparing the CCS and OHS microbial sequencing results obtained from In vitro fermentation and animal experiments with the database, metabolic pathways with consistent changes were found, and KO enzyme expression levels were analyzed.

The heat map of the metabolic pathway predicted by KEGG is shown in Figure 4A,B. After in vitro simulation lyses of OHS, the metabolic pathways of Caprolactam degradation and African trypanosomiasis were significantly weakened, while the eight metabolic pathways, including ethylbenzene degradation and N–Glycan biosystemthesis, were significantly enhanced. According to the results obtained from in vivo lysis, the levels of seven pathways including plant hormone signal transcription and cyanoamino acid metabolism were significantly downregulated, while the phenylalanine metabolism, hypertrophic cardiomypathy (HCM), and cysteine and methionine metabolism pathways were strengthened. By comparing the results of the two groups, we found that the results of phenylalanine metabolism and cysteine and methionine metabolism are consistent. Therefore, based on common predictions in vivo and in vitro, OHS is highly likely to enhance these two metabolic pathways. It should be pointed out that both phenylalanine metabolism and cysteine and methionine metabolism are essential amino acid metabolism pathways. The enhancement of phenylalanine metabolism can relieve the damage of hypoxic cells [45], while cysteine and methionine, as sulfur-containing amino acids, have certain antioxidant capacity [46], so the enhancement of these two metabolic pathways is obviously beneficial to health.

Figure 4C and Figure 4D show some of the phenylalanine metabolism and cysteine and methionine metabolism, respectively, and are marked with significantly changed KO metabolase genes (*p* < 0.05). In the phenylalanine metabolism pathway, the gene expression of the aromatic-L-amino-acid [EC:4.1.1.28] and D-alanine transaminase [EC:2.6.1.21] is upregulated, and the gene expression of the phenylalanine dehydrogenase [EC:1.4.1.20] is downregulated, which is most likely to have a certain effect on the levels of L-phenylalanine and D-phenylalanine. In the cysteine and methionine metabolism pathway, the expression of cystathionine gamma-lyase [EC:4.4.1.1], S-ribosylhomocysteine lyase [EC:4.4.1.21] and S-adenosylmethionine synthetase [EC:2.5.1.6] genes are upregulated, and cystathionine gamma-lyase [EC:4.4.1.1] plays a key role in the liver metabolization of H2S and cysteine [47], and its expression improves to have a certain protective effect on liver injury. Meanwhile, the expression of cystathionine beta-synthase [EC:4.2.1.22] and adenosylhomocysteine nucleosidase [EC:3.2.2.9] may have a certain effect on the L-Homocysteine content in the metabolic pathway.

### 3.4. Validation of Microbial Metabolites

Metabolomics can be used to study the effect of OHS on metabolite changes during in vitro fermentation and in vivo metabolism by qualitative and quantitative analysis of small-molecule metabolites in fermentation broth and in mice. In this study, metabolomics analysis was first performed on the metabolites of OHS after in vitro fermentation. In terms of β-diversity (Figure 5A), the results of principal component analysis (PCA) revealed significant differences between OHS and CCS, indicating that OHS had a certain effect on the metabolite composition. The results of metabolic pathway analysis (Figure 5C) showed that 15 metabolic pathways, including Arginine biosynthesis, showed significant enhancement, among which the enhancement of phenylalanine metabolism was the most obvious, which was consistent with the previous prediction of KEGG. The results of the volcano diagram (Figure 5E) further showed that the metabolites of OHS after in vitro fermentation were up-regulated or down-regulated to varying degrees, among which tanrocholic acid and lactic acid were significantly up-regulated and L-homocystine and phenylalanine were significantly down-regulated, which was consistent with the previous prediction and the results of the metabolic pathway analysis. The significant down-regulation of L-homocystine may be due to the enhancement of cysteine and methionine metabolism, while the significant down-regulation of phenylalanine may be due to the enhancement of phenylalanine metabolism. Screening of differential metabolism substances features, visualized as heatmaps in Figure 5G (VIP > 1, *p* < 0.05, FC > 2), showed that OHS significantly modulated tanrocholic acid, lactic acid, L-homocystine, glycoursodeoxycholic acid, phenylalanine, 12,13-DHOME, and other metabolites, in which the content of tanrocholic acid and lactic acid was significantly elevated, which coincided with the results of the volcano plot.

Follow-up further increased the reliability of the study by probing the in vivo fermentation in mice, and the results of β-diversity (Figure 5B) showed that although there were a few similarities in metabolites between OHS and CCS, the differences were still more significant, further confirming the effect of the OHS on the metabolites. The results of metabolic pathway analysis also showed that Phenylalanine metabolism was significantly up-regulated, which was consistent with the results of previous metabolic analysis during in vitro fermentation. The results of the volcano plot (Figure 5D) further demonstrated that the metabolites of oxidized hydroxypropyl were up-regulated or down-regulated to varying degrees by the in vivo fermentation, where pantothenic acid, chenodeoxycholic acid glycine conjugate, deoxycholic acid glycine conjugate, D-phenylalanine, and L-homocystine were significantly down-regulated (*p* < 0.05) as differential metabolites, where the trends of D-phenylalanine and L-homocystine were consistent with the results of in vitro fermentation metabolite analysis. Similarly, the characteristics of the differential metabolites were visualized as a heat map as shown in Figure 5H. The results showed that compared with the CCS group, the OHS mainly reduced pantothenic acid, L-homocystine, D-phenylalanine, deoxycholic acid glycine conjugate, chenodeoxycholic acid glycine conjugate, ciliatime, and phenylalanine content, which is in agreement with the results of the volcano plot. In addition, OHS fermentation increased the content of N-nitrosodiethanol and calcitriol. Among them, both in vivo and in vitro fermentation metabolite analysis results showed a decreasing trend of pantothenic acid and L-homocystine, and at the same time, they were in high agreement with the KEGG prediction results.

Figure 5I,J shows the thermograms of the same metabolites during in vitro fermentation and in vivo fermentation in mice, and the results showed significant changes in 13 metabolites in the oxidized hydroxypropyl group adenine, pantothenic acid, and phenylalanine compared to CCS, while L-homocystine and phenylalanine changes were consistent with the predicted results, which further demonstrated that OHS may affect the metabolism of L-homocystine and phenylalanine from the metabolite point of view. It has been shown that L-homocystine is formed from S-adenosylmethionine via past methylation and adenosine release and that, when the methionine cycle pathway and other pathways are impaired, L-homocysteine accumulates, leading to hyperhomocysteinemia, a biomarker of cardiovascular disease, neurological/psychiatric disorders, and cancer. Therefore, reduced levels of L-homocystine can help reduce the risk of cardiovascular disease [48]. In addition, studies have shown that diets containing high levels of Phenylalanine may induce phenylketonuria, and a reduction in Phenylalanine has positive implications for people with phenylketonuria.

## 4. Conclusions

In this paper, the anti-digestive properties and intestinal prebiotic effects of OHS and HPS were investigated in depth. The RS content of HPS and OHS was elevated compared to the CCS, and the intestinal prebiotic effects of both HPS and OHS were found to be significantly altered by microbial fermentation in vivo. Multiple correlation analysis in vivo and in vitro revealed that beneficial bacteria such as *Bacteroides_uniformis* and *Parabacteroides_distasonis* increased during fermentation in vivo and in vitro. *Bacteroides_uniformis* was able to alleviate colitis, *Parabacteroides_distasonis* was able to exert intestinal epithelial activity, and *Bacteroides_distasonis* was able to exert intestinal epithelial activity. Distasonis was able to exert intestinal epithelial cell monolayer enhancement and strengthen the intestinal barrier. In this experiment, the preferred OHS was selected for further metabolic analysis, and KEGG prediction showed that OHS enhanced cysteine and methionine metabolism and phenylalanine metabolism, which may affect the content of metabolites in these two metabolic pathways. The final metabolic results verified both in vivo and in vitro that the levels of L-homocystine and phenylalanine were significantly down-regulated, with the reduction of L-homocystine favoring the reduction of cardiovascular and cerebral vascular risks and the reduction of phenylalanine having a positive effect on the alleviation of symptoms in patients with phenylketonuria. Our study provides insight into the structure and properties of hydroxypropyl and hydroxypropyl oxide and demonstrates that hydroxypropyl oxide has intestinal prebiotic effects that can provide the basis for the improvement of related diseases.

## Figures and Tables

**Figure 1 foods-14-02217-f001:**
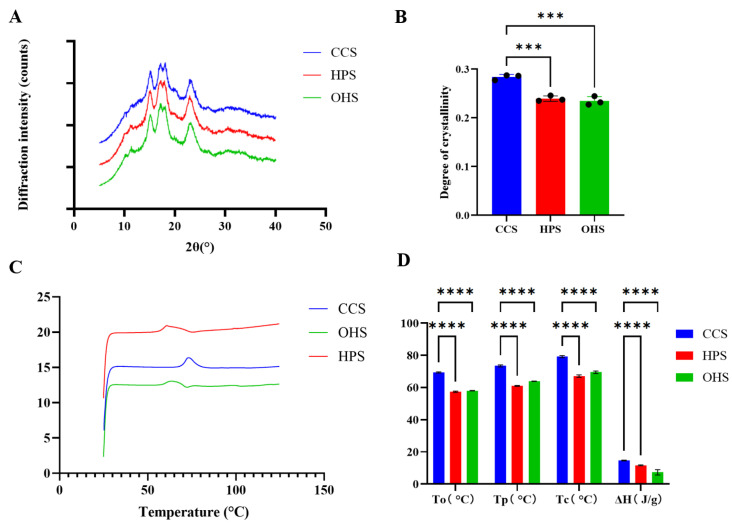
(**A**) XRD pattern of CCS, HPS, OHS; (**B**) Significance analysis plot of the crystallinity of three starches; (**C**) DSC pattern of CCS, HPS, OHS; (**D**) Significance analysis plot of T_0_, T_p_, T_C_, ΔH. Significance levels are:, *** *p* < 0.001, **** *p* < 0.0001.

**Figure 2 foods-14-02217-f002:**
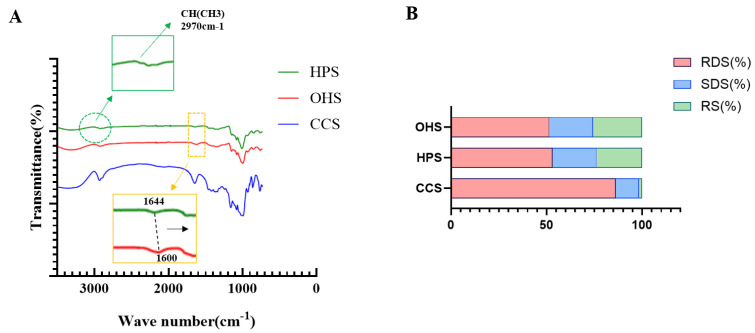
(**A**) FT−IR spectra of CCS, OHS, HPS; (**B**) RDS, SDS, and RS contents of CCS, OHS, HPS.

**Figure 3 foods-14-02217-f003:**
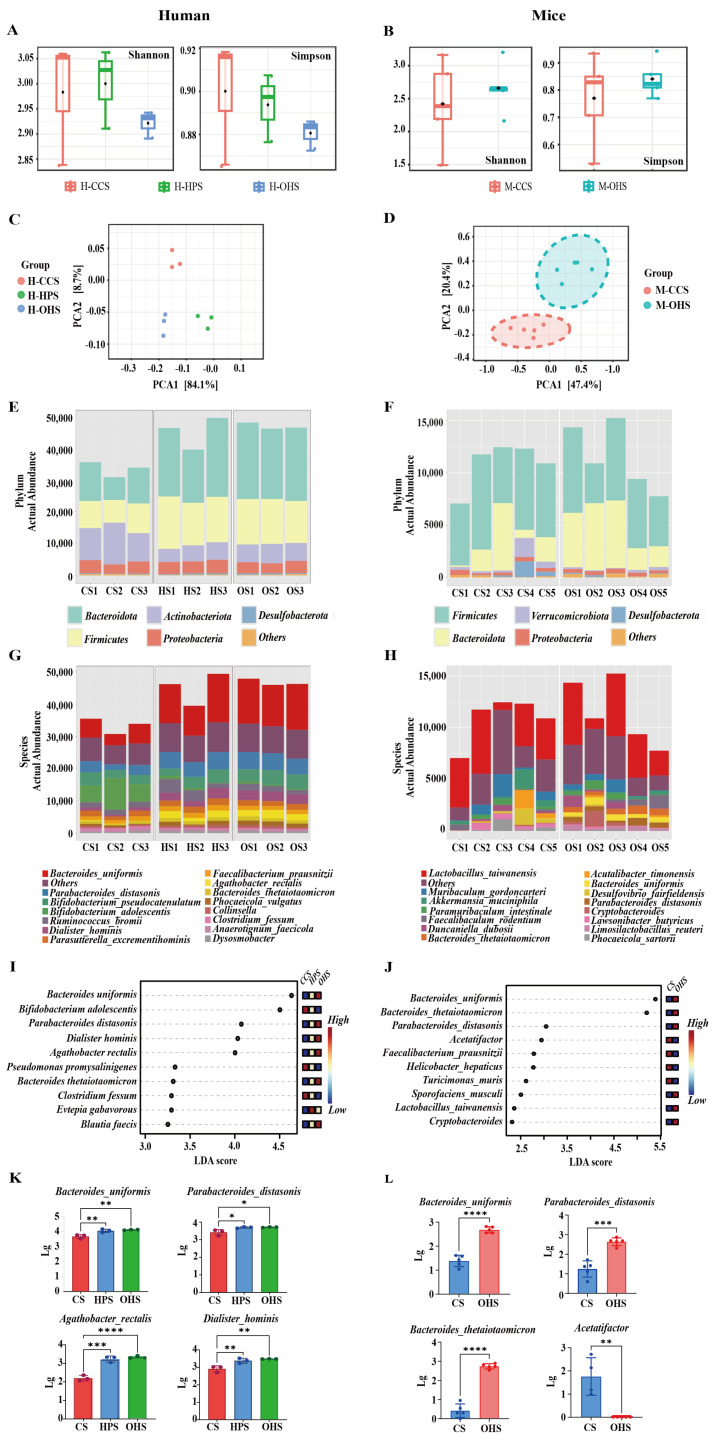
(**A**,**B**) Analysis of microbial α-diversity; (**C**,**D**) β-diversity during in vitro and in vivo fermentation; Variations and differences at the phylum level (**E**,**F**) and species level (**G**,**H**); (**I**–**L**) Characteristic microorganisms and their significance analyses at species level. Significance levels are: * *p* < 0.05; ** *p* < 0.01, *** *p* < 0.001, **** *p* < 0.0001.

**Figure 4 foods-14-02217-f004:**
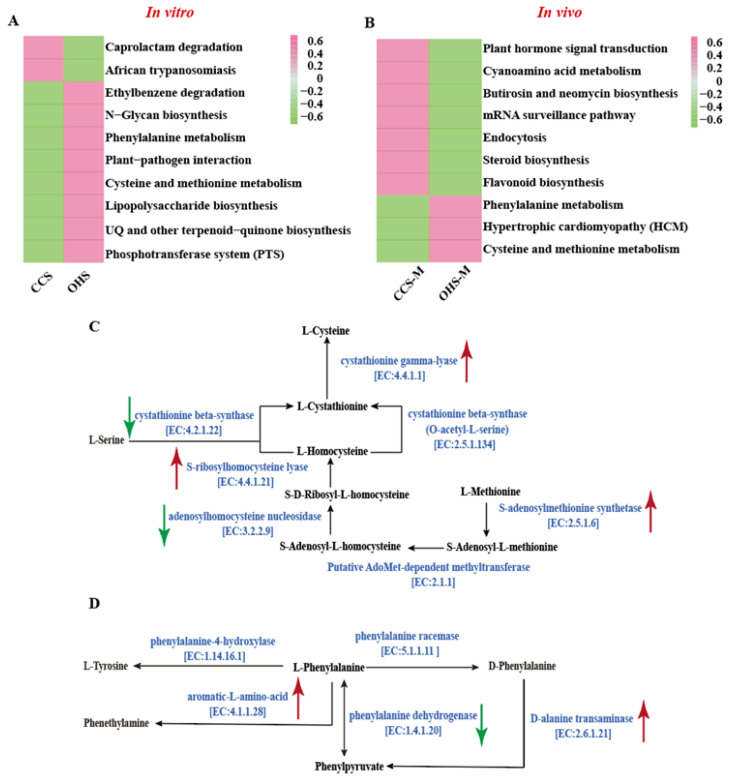
Changes in KEGG pathways in vitro and in vivo (**A**,**B**); Effects of OHS on phenylalanine metabolism pathway (**C**,**D**); Red line: upregulate; Green line: downregulate.

**Figure 5 foods-14-02217-f005:**
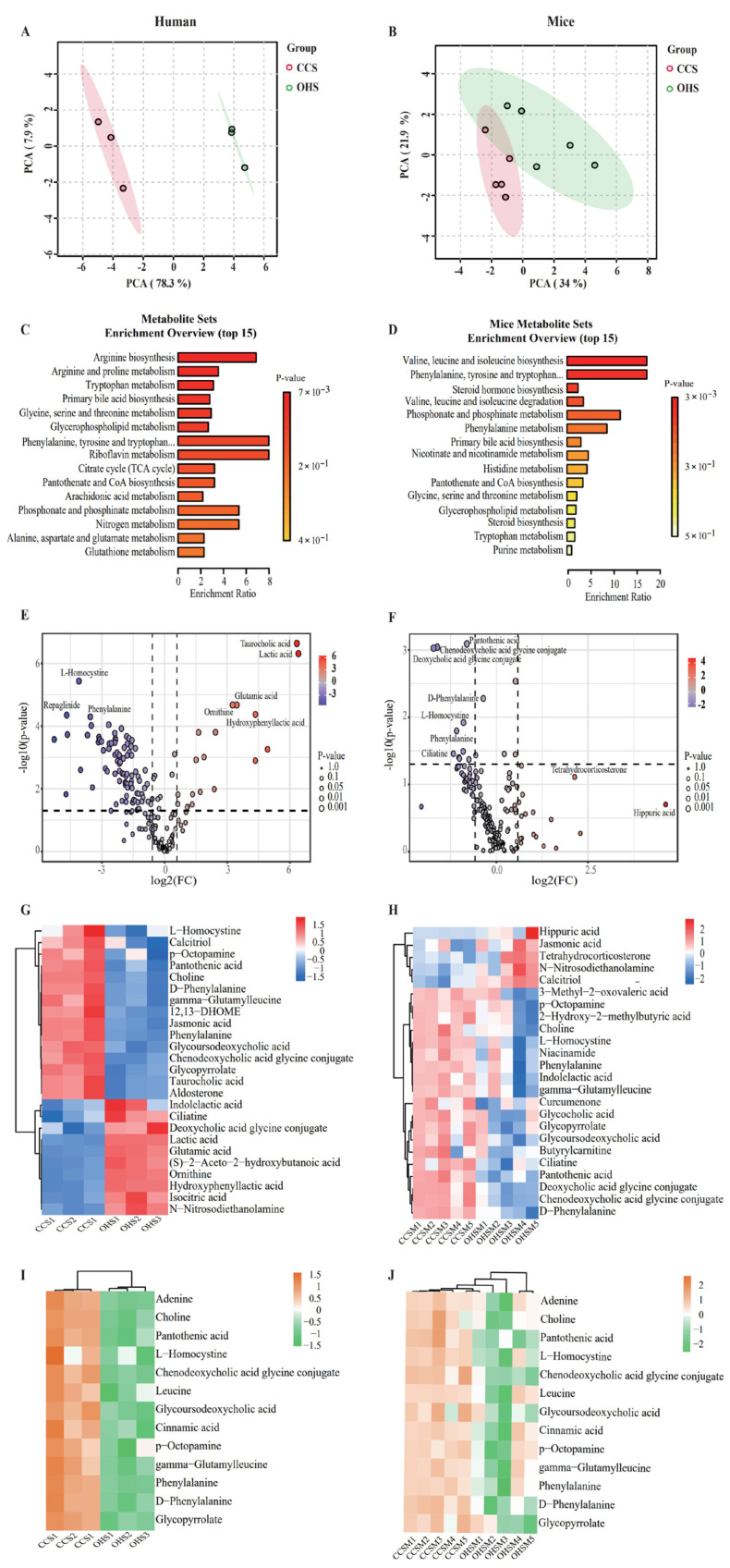
(**A**) Effect of OHS on metabolite β diversity in vivo; (**B**) Effect of OHS on metabolite β diversity in mice; (**C**) Diversity enrichment analysis in vivo; (**D**) Diversity enrichment analysis in mice; (**E**) Volcano plot of metabolite in vivo; (**F**) Volcano map of metabolite in mice; (**G**) Heatmap of differential metabolites in vivo; (**H**) Heatmap of differential metabolites in vivo; (**I**) Co-altered metabolites in vitro; (**J**) Co-altered metabolites in vivo.

**Table 1 foods-14-02217-t001:** Carboxyl content of HPS and OHS.

Sample	Carboxyl Contents (%)
HPS	0.00
OHS	0.56 ± 0.03

## Data Availability

The original contributions presented in the study are included in the article, further inquiries can be directed to the corresponding author.

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
