# Peer review of "Prebiotic Effect of Oxidized Hydroxypropyl Starch via In Vitro and In Vivo"

_foods, 2025, doi:10.3390/foods14132217_

Round 1
Reviewer 1 Report
Comments and Suggestions for Authors
At the outset, the paper deviates significantly from its intended focus. The title or claim stating "Probiotic effect... starch" is inaccurate and misleading, starch is not a microorganism, and therefore cannot exert a probiotic effect. The correct terminology in this context should be "Prebiotic effect", as starch can influence gut microbiota by serving as a substrate, not by acting as a living organism.
Furthermore, the majority of the manuscript centers on the functional properties and characteristics of starch, rather than its interactions with probiotics or its role in gut microbial modulation. This shift in focus undermines the coherence of the paper.
Please ensure a thorough and precise revision of the manuscript to avoid such fundamental conceptual errors.
Author Response
Comments 1:[At the outset, the paper deviates significantly from its intended focus. The title or claim stating "Probiotic effect... starch" is inaccurate and misleading, starch is not a microorganism, and therefore cannot exert a probiotic effect. The correct terminology in this context should be "Prebiotic effect", as starch can influence gut microbiota by serving as a substrate, not by acting as a living organism.]
Response 1:Thank you for pointing this out. We agree with this comment. There is indeed a significant difference between probiotic effects and prebiotic effects. We have therefore read the full article in detail and corrected all references to the gut probiotic effects of resistant starch and changed them to ‘prebiotic effects’. In the revised manuscript, the changes can be found at page 1, lines 2/13/23/40, page 2, lines 41/42/53/55, page 19, lines 630/632/648, which we have marked in red.
Comments 2:[Furthermore, the majority of the manuscript centers on the functional properties and characteristics of starch, rather than its interactions with probiotics or its role in gut microbial modulation. This shift in focus undermines the coherence of the paper.]
Response 2:Thank you very much for pointing out this question, and we agree with you. We retained three methods that are important for "structurally validating the successful preparation of etherified starch": "Fourier infrared spectroscopy verification", "XRD crystal structure determination" and "DSC". Fourier infrared spectroscopy can prove that this etherification modification is successful in the change in the intensity of chemical groups. XRD assists to prove the change in its crystalline structure, and DSC reacts from the thermodynamic properties to the changes in the properties after etherification modification. However, we deleted the polarization microscopy measurement and particle size determination, because these two verification methods can only verify that the starch has been modified, but cannot further prove that the experiment has been modified by hydroxypropylation and oxidized hydroxypropylation, which is not very helpful in proving that our specific etherification modification is successful. In the revised manuscript, the changes can be found at page 7, lines 276-318, and page 10, lines 361-375, as well as at page 3, lines 115-129, and page 5, lines 159-165, of the corresponding Materials and Methods. Similarly, the title of the chart has been amended on page 9, line 332 of the article.
Reviewer 2 Report
Comments and Suggestions for Authors
This is a fascinating work to investigate the changes in the properties of hydroxypropyl starch (HPS) and oxidized hydroxypropyl starch (OHS) relative to common corn starch (CCS), and their probiotic effects on the intestinal tract. However, I have major concerns concerning the analysis of the investigation.
First at all, authors need to have an exhaustive edition. This work is too extensive, leading to one getting lost in the details. As the title indicates, the main objective of this work is the probiotic effect of OHS and not their chemical and physicochemical characterization. In this regard, I recommend delimiting the discussions, mainly the figures, which are not entirely distinguishable in some cases.
In Figure 2, the spectra alone would prove that OHS are efficiently formed, and the graphs could be placed in a supplementary material section.
The same applies to the other figures.
It is important to emphasize that this work is aimed at the functional biological aspect and not specifically at the characterization of the modified materials.
In line 363, authors claim that “In addition, from the infrared map of OHS, it can be found that the bending vibrational absorption peak of OHS near 1640 cm-1 is shifted toward 1600 cm-1 compared to HPS”, is this to conclude the formation of carboxylic acids in the starch? Or what is the reason for this observation?
Author Response
Comments 1:[First at all, authors need to have an exhaustive edition. This work is too extensive, leading to one getting lost in the details. As the title indicates, the main objective of this work is the probiotic effect of OHS and not their chemical and physicochemical characterization. In this regard, I recommend delimiting the discussions, mainly the figures, which are not entirely distinguishable in some cases.
In Figure 2, the spectra alone would prove that OHS are efficiently formed, and the graphs could be placed in a supplementary material section.
The same applies to the other figures.
It is important to emphasize that this work is aimed at the functional biological aspect and not specifically at the characterization of the modified materials.]
Response 1:Thanks for your reply, we agree with you that “this work is directed at functional biology, not specifically targeting the characterization of modified materials”. In addition, you also mentioned that "OHS can be proved efficiently with spectroscopy", so we streamlined and optimized the structural verification analysis. As you said, Fourier infrared spectroscopy can accurately verify that etherification modification is successful by changing the intensity of chemical groups. However, we also retain two other important methods, "XRD crystal structure determination" and "DSC". This is because, although FTIR can detect the introduction of new functional groups, it has limited sensitivity to structural changes, especially in the case of low substitution of oxidized hydroxypropyl, the spectral changes are not significant, and crystallinity and thermal characteristics are required to further characterize structural changes, as evidenced by reference below. We think these two methods are useful for our verification of starch modification, and that retaining only the FTIR spectra would be less convincing. In addition, we deleted polarization microscopy measurement and particle size determination, because these two verification methods can only verify whether the starch has been modified, but cannot further prove whether the experiment has been modified by hydroxypropylation and oxidized hydroxypropylation, which is not very helpful for the proof of the structure of OHS and HPS prepared by us.
In addition, we put all "structural representations" in Chapter 3.1 of the article and "prebiotic effect verification" in Chapter 3.2 of the article structure so that we can clearly define the focus of the article expression. And we modified and optimized the titles of all Figures to clarify the meanings expressed by the Figures for structural characterization and prebiotic effects verification, respectively.
We sincerely appreciate your suggestion regarding moving some figures to the supplementary materials. However, we believe that the included figures provide essential data that supports the core findings and facilitate the reader’s understanding of the main text. Relocating them may disrupt the logical flow and weaken the clarity of the presented results.
Therefore, we would kindly prefer to retain them in the main manuscript, and we hope the reviewer agrees that their inclusion helps strengthen the overall presentation and interpretation of our study. In the revised manuscript, the changes can be found at lines 115-129 on page 3, in the corresponding Materials and Methods on page 5, lines 159-165, and in the main text on page 7, lines 276-318, and page 10, lines 361-375. Similarly, the title of the chart has been corrected, on page 9, line 332, and in subsequent articles in the text.
Reference:Warren, F. J., Gidley, M. J., & Flanagan, B. M. (2016). Infrared spectroscopy as a tool to characterise starch ordered structure—a joint FTIR–ATR, NMR, XRD and DSC study. Carbohydrate polymers, 139, 35-42.
Comments 2: [In line 363, authors claim that “In addition, from the infrared map of OHS, it can be found that the bending vibrational absorption peak of OHS near 1640 cm-1 is shifted toward 1600 cm-1 compared to HPS”, is this to conclude the formation of carboxylic acids in the starch? Or what is the reason for this observation?]
Response 2: Thank you very much for your suggestions. We reviewed two papers for support, which mentioned that when starch is oxidized, the bending vibrational absorption peak at 1640 cm-1 moves to 1600 cm-1 and can be used to prove that starch oxidation is successful. I've included both below and added a reference to the article (there was only one).
Reference Article 1:Oxidation of cornstarch using oxygen as oxidant without catalyst.
Reference:Ye, S., Qiu-hua, W., Xue-Chun, X., Wen-yong, J., Shu-Cai, G., & Hai-Feng, Z. (2011). Oxidation of cornstarch using oxygen as oxidant without catalyst. LWT-Food Science and technology, 44(1), 139-144.
Reference Article 2:Influence of carboxyl content on the rheological properties and printability of oxidized starch for 3D printing applications.
Reference:Zhang, Y., Lv, J., Qiu, Z., & Chen, L. (2025). Influence of carboxyl content on the rheological properties and printability of oxidized starch for 3D printing applications. International Journal of Biological Macromolecules, 289, 138794.
Reviewer 3 Report
Comments and Suggestions for Authors
- Ensure that all experimental protocols are described in enough detail to allow reproducibility.
- In the introduction section: The research gap could be more explicitly stated in the last paragraph.
- Add a clear statement of the study’s objective at the end of the introduction.
- In Materials and Methods section: Specify concentrations, durations, equipment models, and settings (e.g., for ultrasonic treatments or chromatographic analysis).
- Include manufacturer details in parentheses consistently.
Author Response
Comments 1:Ensure that all experimental protocols are described in enough detail to allow reproducibility. Response 1:Thank you very much for your suggestion. We fully agreed. We have added a lot of details to the Materials and Methods section of the text, such as: concentration of reagents, experimental details, manufacturers of products and equipment, etc. For example,in ” Analysis of metabolites by GC-MS and LC-MS measurements” section, we added equipment models of GC-MS and LC-MS(Page 6,paragraph 3,line 243~244). In addition, we also added equipment models of the specific operations of GC-MS and LG-MS, including the durations, settings(Page 6,paragraph 3,line 245~251).
Adding details to materials and methods is essential, and by adding details we can make our experiments reproducible |
Comments 2: In the introduction section: The research gap could be more explicitly stated in the last paragraph. |
Response 2: Thank you for pointing this out. We agree with this comment. Therefore, we have added a paragraph describing the current potential that etherized starch may have and the gaps that exist in current research, in the 3 paragraph on the 2 page of the article, lines 64 ~66. The inclusion of this paragraph gives the reader a clearer picture of the significance of this study. |
|
Comments 3: Add a clear statement of the study’s objective at the end of the introduction. |
Response 3: Thank you very much for your suggestion. We Agree. We have listed the research objectives of this paper in bullet points, rather than in one paragraph as originally written. The modification is in the third paragraph on the second page of the article, lines 66to 71. In addition, we have revised a paragraph for a brief description of the work done in this study, echoing the objective of the study. The modification is in the third paragraph on the second page of the article, lines 75 to 80. Listing the significance of this article in points allows the reader to read this article more purposefully and also makes it more professional, so this is very important. Comments 4:In Materials and Methods section: Specify concentrations, durations, equipment models, and settings (e.g., for ultrasonic treatments or chromatographic analysis). Response 4: Thanks for the advice. We agree with your suggestion. We have added the concentrations of Corn starch, anhydrous sodium sulfate and propylene oxide in the “Preparation of hydroxypropyl starch (HPS)” section. (Page 3,paragraph 2,line 93 and line 96)We also added the instrumentation used for pH determination. (Page 3,paragraph 2,line 94~95) In the section “Preparation of oxidized hydroxypropyl starch (OHS)” we added the concentration of hydroxypropyl starch suspension(Page 3,paragraph 3,line 106),and the equipment models for water baths(Page 3,paragraph 3,line 107~108). We have added equipment models for centrifuged, thermostatic water bath shaker and Enzyme-Labeled Instrument to the “In vitro digestion characterization” section(Page 5,paragraph 2,line 182,190~191,198~199) and we also have added amyloglucosidase, aqueous ethanol concentration (Page 5,paragraph 2,line 184 and line 194)and water bath settings to this section(Page 5,paragraph 2,line 184 and line 191). For” Analysis of metabolites by GC-MS and LC-MS measurements” section, we added equipment models of GC-MS and LC-MS(Page 6,paragraph 3,line 243~244). In addition, we also added equipment models of the specific operations of GC-MS and LG-MS, including the durations, settings(Page 6,paragraph 4~5,line 246~251). We have added equipment models for PCR amplification in the “16S rRNA gene and bioinformatics analysis” section (Page7,paragraph1,line259). Adding these can improve the reproducibility of the experiment and show the experimental procedure more clearly. Therefore, it is very important to add these elements. Comments 5: Include manufacturer details in parentheses consistently. Response 5: Thank you for your suggestion, we have added the equipment model number and equipment manufacturer after all the equipment. In” Preparation of hydroxypropyl starch (HPS)”section,we add the equipment model number and equipment manufacturer for precision pH meter in page 3,paragraph 2,line94~95. We have added the equipment model number and equipment manufacturer for centrifuged, thermostatic water bath shaker and Enzyme-Labeled Instrument to the “In vitro digestion characterization” section(Page 5,paragraph 2,line 182,190~191,198~199) In the “Analysis of metabolites by GC-MS and LC-MS measurements”section,we added the equipment model number and equipment manufacturer for GC-MS and GC-LS in page 6,paragraph 3,line 243~244)
|
Round 2
Reviewer 1 Report
Comments and Suggestions for Authors
The authors have addressed all the queries and suggestions, and have incorporated the necessary changes in the revised manuscript. Therefore, the paper is now suitable for publication.